# Lichen Sclerosus: A Current Landscape of Autoimmune and Genetic Interplay

**DOI:** 10.3390/diagnostics12123070

**Published:** 2022-12-06

**Authors:** Noritaka Oyama, Minoru Hasegawa

**Affiliations:** Department of Dermatology, Faculty of Medical Sciences, University of Fukui, 23-3 Matsuoka-Shimoaizuki, Eiheiji, Fukui 910-1193, Japan

**Keywords:** lichen sclerosus, extracellular matrix protein 1, lipoid proteinosis, basement membrane zone, laminin-332, collagen IV, collagen VII, glycosaminoglycan

## Abstract

Lichen sclerosus (LS) is an acquired chronic inflammatory dermatosis predominantly affecting the anogenital area with recalcitrant itching and soreness. Progressive or persistent LS may cause urinary and sexual disturbances and an increased risk of local skin malignancy with a prevalence of up to 11%. Investigations on lipoid proteinosis, an autosomal recessive genodermatosis caused by loss-of-function mutations in the extracellular matrix protein 1 (*ECM1*) gene, led to the discovery of a humoral autoimmune response to the identical molecule in LS, providing evidence for an autoimmune and genetic counterpart targeting ECM1. This paper provides an overview of the fundamental importance and current issue of better understanding the immunopathology attributed to ECM1 in LS. Furthermore, we highlight the pleiotropic action of ECM1 in homeostatic and structural maintenance of skin biology as well as in a variety of human disorders possibly associated with impaired or gained ECM1 function, including the inflammatory bowel disease ulcerative colitis, Th2 cell-dependent airway allergies, T-cell and B-cell activation, and the demyelinating central nervous system disease multiple sclerosis, to facilitate sharing the concept as a plausible therapeutic target of this attractive molecule.

## 1. Introduction

Lichen sclerosus (LS), also known as ‘lichen sclerosus et atrophicus’, ‘balanitis xerotica obliterans’, ’kraurosis vulvae’, or ‘hypoplastic dystrophy’, is an acquired chronic inflammatory disease that primarily affects the skin and mucous membranes, with a high occurrence in the anogenital area [1,2,3]. LS represents one of the most common referrals for pruritis and structural alteration in the vulva [1,4]. The predilection sites of the disease often cause serious urinary and sexual dysfunction, including dyspareunia and psychological impairments. In addition, LS has been associated with an increased risk of malignancy, mostly squamous cell carcinoma, in long-standing lesions in both sexes [5,6,7,8]. For therapeutic remedies, the topical application of potent corticosteroids is a primary mainstay [4,9,10]. Treatment options for refractories to the standard treatment regimens include systemic or local immunosuppressants (e.g., oral or topical calcineurin inhibitors), retinoids, phototherapy, and photodynamic therapy [11,12,13,14,15]. The mechanisms of action of these agents imply the possible involvement of an immunological disturbance in LS.

Although the pathogenesis of LS has yet to be elucidated, a series of etiological and epidemiological studies have suggested a possible genetic susceptibility and an autoimmune basis for the disease. For example, LS has a higher familial predisposition [16,17]; of 1052 individual cases with LS, 126 (~12%) had family histories. In addition, there have been increasing reports of monozygotic and dizygotic twins with LS. Genetic assessment of reliable numbers of LS cohorts identified a high association with the presence of particular human leukocyte antigens (HLAs) and haplotypes [18,19,20], such as DQ7, DR12, DRB1*12, and DRB1*13. Clinically, LS tends to coincide with various promiscuous autoimmune diseases with serum autoantibodies, such as morphea, Hashimoto’s thyroiditis, rheumatoid arthritis, pernicious anemia, type I diabetes mellitus, alopecia areata, vitiligo, morphea, and immuno-bullous mucocutaneous diseases [21,22,23,24,25,26,27]. Of these, serum anti-thyroid antibodies were highly detectable in female LS (11–12%), compared to normal females. A series of clinical evidence has gradually suggested a scenario for the possible involvement of autoimmune imbalance, particularly autoimmune response to skin antigen(s), in LS.

More critically, loss-of-function mutations in the extracellular matrix protein 1 (*ECM1*) gene were demonstrated in an autosomal recessive genodermatosis referred to as lipoid proteinosis (LiP), a counterpart disease that is a similar skin pathology to LS [28]. This clinical rationale has led to the identification of humoral autoimmunity to ECM1 protein in patients with female anogenital LS [29], which was also applicable in the immunopathogenesis of male penile LS [2]. Furthermore, the similar clinicopathology between LS and LiP has allowed for considerable progress in addressing the in vivo biological function of this attractive molecule in animal model studies using a passive transfer of anti-ECM1 antibodies, and gene knockout and transgenic systems targeted for ECM1 in mice and zebrafish [30,31,32,33]. In addition, recent studies have demonstrated the role of ECM1 in the genetic predisposition to the inflammatory bowel disease (IBD) ulcerative colitis [34,35], the acquisition of immune tolerance and allergic responses via particular T-cell subsets such as CD4+CD25+ regulatory T cells and Th2 cells [31,36], and activation of abundant B-cell biology [33]. Thus, the biology of ECM1 positions it as an attractive core candidate to interconnect disease-specific pathophysiology.

## 2. Clinical Characteristics of LS

### 2.1. Clinical Features

LS mostly affects the anogenital skin and mucous membranes in the vast majority of cases (85–98%), typically female [1,2,3]. In male patients, the site-specific predisposition has been associated with a high occurrence rate of LS on normal penile skin grafts whose original diseases were unrelated to LS [37], but this is not a fate of the original skin sites. Without regard to gender, the typical clinical picture includes erosions as well as whitish-pale, indurative polygonal (porcelain-like) papules and plaques, which later change to atrophic scarring (Figure 1). An itching sensation, among various symptoms, is primary and inevitable [38]. The lesion may account for the bullous and hemorrhagic appearance, but it causes a potential difficulty in arriving at an accurate diagnosis [39,40]. Several reports for extragenital LS cases have suggested a variety of affected skin sites including the head, neck, scalp, palms and soles, periorbital area, tongue, lip, and peristoma skin [41] (Figure 2). Interestingly, extragenital LS, unlike genital disease, seldom develops local skin malignancies, giving way to the assumption that genital circumstances can be a primary extrinsic factor in the development of LS. The proposed candidate may include occasional herpes virus infection, urine and/or fecal irritation, constitutive colonization of gram-negative bacteria, or any combination of these.

### 2.2. Epidemiology and Etiology

Results from epidemiological studies indicated that LS was probably underreported and may have had a prevalence of approximately 0.1–0.3%, with a male-to-female ratio of 1:10 [42,43]. LS can develop at any age; however, a bimodal peak has been found in prepubertal girls and postmenopausal women as well as in middle-aged men [2,42,44,45,46]. A retrospective analysis using a large number (n = 411) of uncharacterized penile dermatoses revealed that ~10% of the foreskin biopsy samples showed the typical LS pathology [47]. Therefore, LS may occur far more frequently than previously expected, although the gender bias remains unchanged (~3% in females and >0.07% in males) [46,48]. The significant sex ratio and age bimodality indicates a possible association with hormonal imbalance, particularly estrogen deficiency, but hormone replacement therapy neither improves existing disease nor provides any protective effects against the disease. Persistent LS may result in severe scarring, which can lead to impairments in micturition and sexual activity, urethral stricture, and associated physical morbidities [49,50], and more importantly an increased risk of malignancy, particularly non-differentiated squamous cell carcinoma (4.7–10.7%) [51,52] (Figure 1).

### 2.3. Histopathological Features

Typical LS pathology displays a thickening epidermis with hyperkeratosis and follicular plugging, particularly in the early clinical stage, consequently followed by atrophic flattening of the epidermal rete ridges (Figure 3). The dermis underneath undergoes zonal hyalinization, intermingled with amorphous eosinophilic materials, homogeneous collagen bundles, and telangiectasia. A band-like infiltration of inflammatory cells may be present along with and/or separate from the hyalinizing dermis, which becomes sparser and more focal during the clinical course. However, each of these pathological findings often coexists at differing frequencies and degrees. Skin biopsies may therefore provide inconsistent pictures and result in difficulty differentiating other mimicking dermatoses affected in the anogenital area, resulting in the dilemma of diagnostic inaccuracy and delay [53,54].

## 3. Molecular Characteristics of ECM1: A Secretory Glycoprotein

### 3.1. Historical Background for the Discovery of the ECM1 Gene: What It Means

The human *ECM1* gene was first isolated in 1997 and was mapped to chromosome 1q21.2, located centromerically to the gene cluster termed epidermal differentiation complex (EDC) [55,56]. Comparing the plane structure to the previously discovered mouse *Ecm1* gene in 1994 [57], the human counterpart represents one exon fewer than the mouse gene; the sequence is homologous to the sixth shortest mouse exon [55]. The upstream regulatory sequences of the human gene contain putative binding sites for various major transcription factors, such as GATA, Sp1, AP-1, and ETS family members, all of which, except for the potential GATA-binding motifs, are highly conserved with the equivalent portion of the mouse *Ecm1* gene [58]. The *ECM* gene is highly conserved and expressed in most eukaryotes and their various cell types [59], supporting the concept of potential significance in the evolutionary process, as was the discovery of a genetic disease caused by mutations in this gene, lipoid proteinosis [28].

### 3.2. Gene Structure and Variants

The human *ECM1* gene is located on chromosome 1q21.2 and encodes four splice variants, ECM1a–d [55,56]. ECM1a (1.8-kb, 540 amino acids) comprises 10 exons, whereas ECM1b (1.4-kb, 415 amino acids) only lacks exon 7. ECM1c (1.85-kb, 559 amino acids) contains an additional exon (5a) within intron 5 of ECM1a. These three major variants show widespread and differential expression patterns in human tissues [58,60]. For example, ECM1a is ubiquitously expressed in major organs including the skin, liver, intestine, lung, ovary, prostate, testis, skeletal muscle, pancreas, and kidney, with the greatest expression levels observed in the placenta and heart. ECM1b expression seems to be restricted to the tonsils and epidermal keratinocytes, whereas the tissue/cell type-specific expression of ECM1c remains to be identified [61]. ECM1d comprises a minimum splicing variant, with an out-of-frame insertion of 71 nucleotides at the 5′ end of exon 2, resulting in a truncated protein of 57 amino acids [62] and an enigmatic biological significance. In skin, ECM1a was expressed in the epidermal basal layer, dermal blood vessels, outer root sheath of hair follicles, sebaceous lobules, and sweat gland epithelia, whereas ECM1b was localized to the suprabasal layers of the epidermis [60,61,63]. Thus, in vivo associations between each of the ECM1 splicing variants and skin biology are gradually forming.

### 3.3. Protein Structure and Function

ECM1 is an 85-kDa-secreted glycoprotein known to play pivotal roles in the structural and homeostatic organization of various skin components through direct binding with various extracellular molecules, such as perlecan, matrix metalloproteinase family members, fibulins, fibrillins, fibronectin, laminin-332, type IV and type VII collagens, cartilage-derived oligomeric matrix protein (COMP), proteoglycans, glycosaminoglycans, phospholipids (particularly phospholipid scramblase 1), and progranulin chondrogenic growth factor (PGRN) [61,64,65,66,67,68,69]. Of note, these molecules co-localize immunohistologically with in vivo ECM1 in human skin. The multifocal interaction between skin structural molecules contributes to the biological significance of ECM1 in epidermal growth and differentiation, basement membrane integrity, angiogenesis, endochondral development, and certain malignancies, as well as in the structural maintenance of the dermis (Figure 4). Most of these biological activities are associated with positive regulation of cell proliferation, migration, and differentiation, resulting in tissue formation and organization; however, negative effects on chondrocyte hypertrophy, matrix mineralization, and endochondral bone formation have been described as well [70,71,72]. ECM1 thus penetrates into the fundamental skin biology via complex organization with surrounding microstructural molecules, and mutations of their corresponding genes are responsible for a hereditary genodermatosis LiP [28].

## 4. Autoimmune Response in LS

### 4.1. Etiological Scenario for Autoimmunity to ECM1 in LS

Although no plausible evidence regarding the local and systemic autoimmune reactions characteristic of LS has been put forth, recent progress on the screening of disease-specific serum autoantibodies in LS is extrapolated from bipolar evidence for possible genetic susceptibility and a humoral autoimmune basis for the disease. Specifically, study findings have identified variable intra-familial cases with LS [16,73], an association with particular HLA class II antigens (DQ7-9, DR11, DR12, and DQ17) [18,20], and a high coexistence of various autoimmune diseases, such as morphea, Hashimoto’s thyroiditis, rheumatoid arthritis, pernicious anemia, type I diabetes mellitus, alopecia areata, vitiligo, bullous pemphigoid, and mucous membrane pemphigoid [21,22,23,24,25,27]. Of these autoantibody- and/or T-cell-driven diseases, anti-thyroid antibodies were highly detectable in female LS patients (11.1–40%) compared with other organ- and tissue-specific autoantibodies. However, the autoantibodies mostly do not correlate with either severity or duration of the corresponding diseases, proving irrelevant as a consequence of LS [74].

A century ago, evidence implicating a humoral autoimmune response in LS was demonstrated in a case where probable LS was induced through the injection of an autologous serum from an affected individual into non-lesional skin [75]. More critically, the skin pathology of LS shares considerable overlap with that of LiP (OMIM 247100), an ECM1-deficient genetic skin disease [28], for example, trauma-induced inflammation (also known as Koebner’s phenomenon) and lesional skin microscopy showing the atrophic epidermis with hyperkeratosis, disruption and duplication of the basement membrane, and hyaline (glassy-like) collagen changes and telangiectasia in the upper dermis (Figure 3). Altogether, this clinicopathological evidence straightforwardly implicates a counterpart disease concept targeting ECM1 in both LS and LiP [76].

### 4.2. Identification of IgG Autoantibodies Reactive with ECM1 in LS

Primary screening using immunoblotting with cultured normal human keratinocyte substrates identified detectable levels of serum IgG-class antibodies to three of the four ECM1 isoforms, namely ECM1a–c in ~70% of female patients with genital LS [29]. The fidelity of the seroreactivity was confirmed using a bacterially generated full-length recombinant ECM1a protein. Thereafter, antigen-specific enzyme-linked immunosorbent assays (ELISAs) utilizing a highly antigenic portion of the recombinant ECM1 protein (359–559 amino acids) optimized the immunoreactivity of serum anti-ECM1 antibodies in 74–80% of female patients with genital LS, with 94% specificity in discriminating LS from other autoimmune diseases and healthy controls [30]. In another cohort study, ECM1 seroreactivity was also detectable in male patients with male penile LS [2]. The update series has further accelerated the serological diagnostic accuracy in individual cases suspicious of LS [39,75]. Irrespective of gender, humoral autoimmunity to ECM1 needs to be considered on the basis of its pathogenic significance in LS.

## 5. Issues that Need to be Addressed to Better Understand ECM1 Autoimmunity in LS

### 5.1. Lack of In Vivo-Bound Anti-ECM1 IgG in the LS Skin

Debate continues regarding the difficulties in detecting the anti-ECM1 autoantibody in LS lesional skin. For example, direct immunofluorescence studies of LS skin have shown no signals [44,77]. In addition, indirect immunofluorescence studies using LS patients’ sera on normal human skin sections also showed negative (with standard dilutions of the sera) or only faintly positive signals along with the lower epidermis and basement membrane. These controversial observations may be attributable to differences in affinity and/or avidity of antigen-specific serum IgG to the in vivo-native ECM1 antigen, because the IgG fraction of affinity purified from LS sera exhibits intense immuno-reactivity in the lower epidermis, and similar observations were made from immuno-labeling with rabbit anti-ECM1 polyclonal antibodies on normal human skin [29] (Figure 4). Based on our recognition, ECM1 is considered a secretory glycoprotein that acts as a biological ‘glue’ to stabilize robust structural proteins [78], such as BPAG I/II, laminin-332, and collagens. One may consider that the flowability and turnover of in vivo ECM1, in part, affect the accessibility and reactivity of the antibody. Considering these technical limitations, the standard immunohistochemical approach using patients’ skin or sera remains less valuable in a routine laboratory workup, and the ELISA system specific for ECM1—or at least immunoblotting using recombinant ECM1 protein—may currently be a preferable tool for a noninvasive and objective serodiagnosis in LS [30].

### 5.2. Difficulty in the Establishment of Mouse Models for LS and LiP

The pathogenic relevance of autoimmunity to ECM1 in LS has been reevaluated by mouse passive-transfer experiments using intra-cutaneous injections of either a rabbit anti-ECM1 polyclonal antibody or an affinity-purified IgG from ECM1 ELISA-positive LS patients’ sera [30]. Both antibodies recognize the regions of the ECM1 protein that are highly homologous between humans and mice, and when injected into the skin, both IgGs were indeed accessible to native mouse ECM1 in vivo, as was identical immunoreactivity to human skin. The mouse skin sites injected either with a rabbit anti-ECM1 polyclonal antibody or an affinity-purified IgG from LS sera exhibited a clinicopathology compatible with the early clinical stage of LS, including erythematous swelling (dermal inflammation) and dilated blood vessels (telangiectasia) for up to two weeks after the initial injection. However, this approach failed to reproduce dermal hyalinosis or scarring, both of which are histological hallmarks of the well-established (late) stage of LS. This incomplete observation raises possible interpretations for how hyalinosis and/or sclerotic events in LS skin are indeed consequences of complex and sensitive events. For example, the sensitivity may depend on the particular genetic background(s) of the mouse strain(s) or the HLA class II haplotypes of the patients [18,20]. In addition, it may be affected by the constitutive impairment of in vivo ECM1 function, e.g., more prolonged exposure to anti-ECM1 antibodies. Considering the clinical aspect that (i) the vast majority of LS affects anogenital skin and (ii) extragenital LS sometimes develops around utero-abdominal fistula [41,79], urinary irritation and/or local infection with gram-negative bacteria may be (a) confounding factor(s).

Additional experiments have explored the fact that conventional/targeted disruptions of the *Ecm1* gene in mice (ECM1^−/−^) render it lethal (surviving for no more than 6–8 weeks) [31,80], strongly suggesting that ECM1 is indispensable for at least early embryonic development. In contrast, LiP patients represent an ECM1-knockout condition in humans who can survive and show no evidence of short life spans or disease-related mortality. In addition to this discrepancy, a systematic disease translational study comparing gene transcriptional responses to inflammatory insults in mice and humans detected disparities in the gene expression profiles between the mouse models and their human counterparts [80]. Species-specific gene regulation may thus represent a challenge to reproducing typical LS pathology in mice.

### 5.3. Establishment of ECM1-Knockdown Human Dermal Fibroblasts

A series of our failures for recapitulating LS and LiP phenotypes in mouse models led us to re-recognize what happens to the impaired ECM1 function in vitro. ECM1 siRNA knockdown in human dermal fibroblasts showed a significant delay of growth and migratory activities, as well as collagen gel contraction, compared to control fibroblasts [69]. Also, the ECM1 knockdown upregulated genes related to structural, fibrogenic, and carcinogenic properties, some of which shared the skin structural molecules that are major binding partners for ECM1, such as laminin-332, and type IV and VII collagens [64,67,69]. All of the binders displayed altered immunolabeling at the basement membrane zone and dermal vessels in the LS lesional skin (Figure 4), where ECM1 is highly expressed. Combining these in vitro data with the ECM1 autoantibody scenario, one may speculate that the antibody-dependent impairment of ECM1 function disrupts a biological interconnection with the in vivo binding partners, resulting in their functional behavior to maintain the structural integrity responsible for the LS pathology (Figure 5).

## 6. Lessens from Novel ECM1 Function in Other Animal Models

### 6.1. Th2 Cell-Dependent Allergic Response in the Airway

DNA microarray assays have disclosed the expression of the *Ecm1* gene in mouse hematopoietic cells, particularly in T cells [81,82], although the transcription levels considerably differ in the T-cell differentiation- and lineage-dependent manner; it was much higher in CD4+ helper T cells and CD4+CD25+ T cells (Tregs) but relatively lower in CD8+ cytotoxic T cells and CD3-negative naïve T cells. Of the CD4+ helper T-cell lineage, the ECM1 expression was almost prone to Th2 cells [31], indicating the possible association between ECM1 and allergic reactions. This scenario was also compounded through the finding that chimeric BALB/c mice transplanted with ECM1-deficient bone marrow cells showed a decrease in inflammatory response in experimentally induced airway allergy. Functional analysis for several T-cell lineages from ECM1-knockout mice exhibited no substantial differences in the proliferation activity, cytokine/chemokine profiles, and polarization of their differentiation, suggesting the direct action of ECM1 in Th2 cell trafficking from lymph nodes into circulation.

In Th2 cells, ECM1 mRNA and protein expression were detectable three days after antigen-dependent engagement of the T-cell receptor. Subsequently, ECM1 can bind with an IL-2 receptor subunit (CD122), but with neither CD25 nor CD132, to inhibit the phosphorylation and activation of the downstream-signaling molecules, such as STAT5, KLF2, and S1P1 [31], resulting in the downregulation of Th2 cell trafficking to the local inflammatory sites.

On the other hand, freshly isolated and activated CD4+ CD25+ Tregs highly express ECM1 transcription [83]. CD4+ CD25+Tregs are well-known to regulate innate and adaptive immune responses, tumor immunity, and a potent anti-inflammatory capacity in autoimmune and chronic inflammatory diseases, such as autoimmune encephalitis, diabetes, thyroiditis, IBDs, and contact skin hypersensitivity [84,85,86,87,88]. Naturally occurring Tregs, the other Treg phenotype that comprises up to 5% of the peripheral CD4+ T-cell pool, have also been shown to express ECM1 [87]. More critically, the ECM1 transcription was significantly increased in naïve T cells by transient transduction of forkhead box P3 (FOXP3), a transcription factor that acts as a master control molecule for the development and function of CD4+CD25+ Tregs in the thymus and periphery [88].

### 6.2. Macrophage Polarization in Inflammatory Bowel Diseases (IBDs)

Genotyping using a reliable number of ulcerative colitis cohorts (n = 905) determined a strong disease susceptibility locus at the ECM1 gene [34]. The foothold identification further accelerated the research activity concerning ECM1-targeted culprit cells in the disease. Mice transplanted with ECM1 knocked-down macrophages, a phenotype unable to polarize towards M1 macrophage, decreased pathological inflammation of colitis in experimental IBD mice [34], raising a direct interpretation of ECM1 function to regulate the IBD-dependent macrophage lineage. However, data from several trials determining ECM1 gene mutation and/or polymorphism in IBD cohorts remains unstable as the genetic signature [88,89], which may be impacted by racial difference. Also, there has been no available evidence of a relationship between ECM1 and the intestinal epithelial barrier, as well as permeability balance and luminal antigen absorption in the intestine interface.

### 6.3. Miscellaneous

#### 6.3.1. Multiple Sclerosis

Multiple sclerosis is a chronic inflammatory condition in the central nervous system characterized by demyelination and axonal damage through Th17-dependent innate immunity. Administration of recombinant ECM1 ameliorated the severity of encephalomyelitis with cerebral demyelination and inflammation, accompanied by a decrease in the Th17 response, in an experimental model for multiple sclerosis [90]. Inversely, in vivo overexpression of ECM1 successfully inhibited encephalomyelitis with Th17 cell activation. The protective action of ECM1 is mediated in part by its direct interaction with av-integrin on dendritic cells, blocking the integrin-mediated activation of TGF-β. Data support a possible engagement in a replacement therapy targeting ECM1.

#### 6.3.2. B Cell Function

Follicular helper T cells (T_FH_) are a subset of the CD4^+^ helper T-cell lineage that enables a variety of B-cell responses, i.e., the formation of germinal centers (GCs), affinity maturation of GC B cells, differentiation of high-affinity antibody-producing plasma cells, and production of memory B cells. All the B cell-specific reactions were impaired by an ECM1 knockout in antigen-immunized mice [33]. Exogenously injected ECM1 into mice infected with influenza virus exhibited a protective immune response via enhancing differentiation towards T_FH_ cells and production of virus-neutralizing antibodies. ECM1 can therefore promote T_FH_ cell differentiation and antibody production, both of which are indispensable for humoral autoimmunity.

## 7. Conclusions

Considerable progress has recently been made in both clinical and animal studies designed to elucidate the in vivo function of ECM1. Novel insights regarding this molecule in the restricted T-cell repertoire, B-cell activation, organ-specific allergic reaction, and genetic susceptibility to ulcerative colitis have been condensed during the last decade. In addition, much attention has been paid to the role of ECM1 in tumor biology, particularly its microenvironment as an alternative to tumor-directed therapy [72,91,92]. Notwithstanding these updates, genetic ablation of ECM1 and passive transfer of ECM1-specific antibodies in mice has yet to fully explain the characteristic pathophysiology in human diseases LiP and LS, respectively. Future studies concerning the establishment of animal models for LS now await sophistication of the overall technical processes.

Clinical observation shows that LiP patients are viable without an inherent susceptibility to any type of cancer, unlike LS patients. ECM1 can therefore be dispensable or at least compensable spatially and temporally for the development of certain organs, including the skin. Future studies will encourage research on the tissue and organ developmental stage-specific significance of ECM1 action.

## Figures and Tables

**Figure 1 diagnostics-12-03070-f001:**
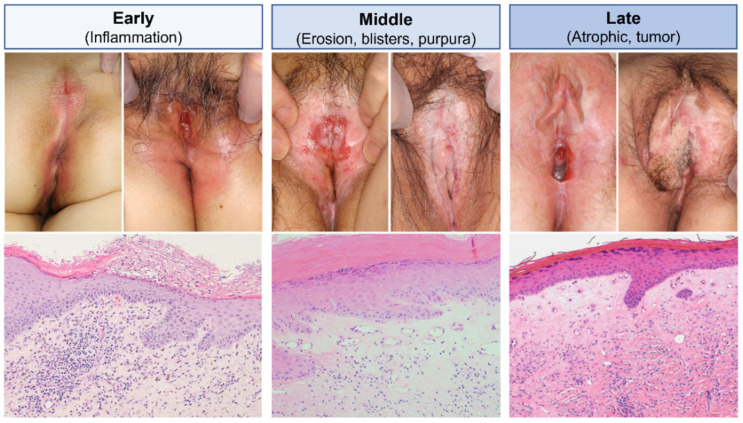
Stage-specific clinical and pathological features of female genital LS. In the early clinical stage of LS (**left columns**), it initiates as erythema and mild erosion, mostly covering the entire peri-vaginal and perianal area. Pathologically, early LS shows parakeratotic scales, irregular epidermal thickening, and intense inflammatory infiltrates in the upper dermis with faint homogenization of dermal collagen bundles. The condition fluctuates and gradually exacerbates into persistent erosions with focal blistering and induration, leading to a whitish-pale appearance in the middle clinical stage (**middle columns**). Note that dermal hyalinosis is more apparent with dilated blood vessels. The chain of these inflammatory events finally results in scarring and irreversible adhesion of external genital parts, and abruptly develops squamous cell carcinoma in the late clinical stage (**right columns**). The preexisting dermal hyalinosis in the skin pathology further extends diffusely, becoming more prone to eosinophilic staining and a thinner color. Dilated blood vessels in the upper dermis look atrophic, narrowing in size with thickening vessel walls.

**Figure 2 diagnostics-12-03070-f002:**
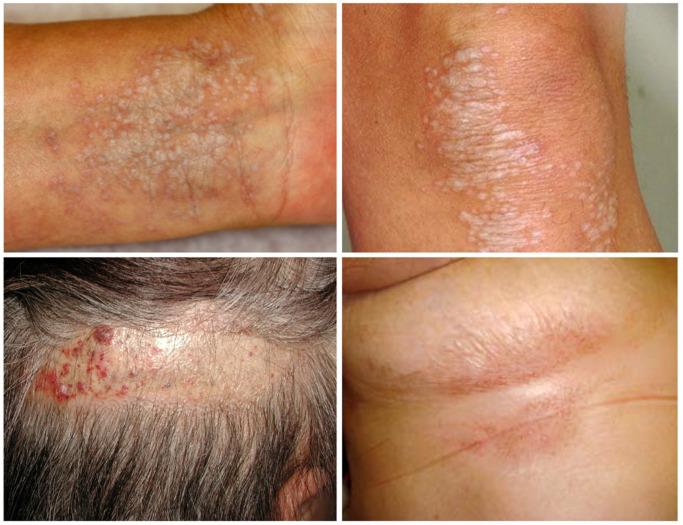
Clinical features of LS developed on the extragenital skin. The extragenital LS occurs at any skin sites, including extremities (**upper left and right**), scalp (**lower left**), and trunk (**lower right**).

**Figure 3 diagnostics-12-03070-f003:**
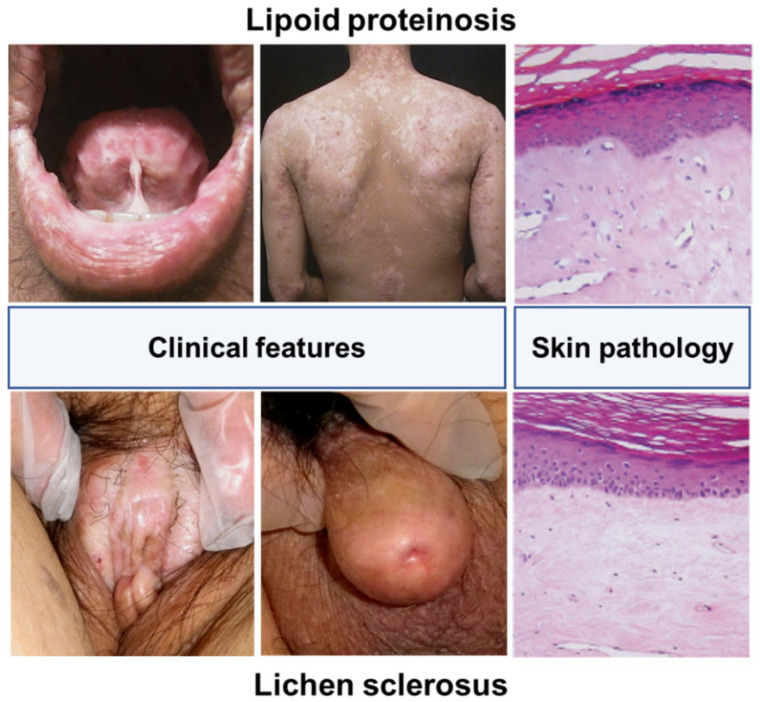
Clinicopathology of lipoid proteinosis (an ECM1-lacking genodermatosis: **upper panels**) and lichen sclerosus (an anti-ECM1 autoantibody-carrying condition in females and males; **left and right in the lower panels, respectively**). Irrespective of a predilection to different skin sites between the two diseases, their skin pathologies display similar features (**right columns**), including packed and parakeratotic hyperkeratosis, epidermal atrophy, and diffuse hyaline changes and dilated blood vessels in the upper dermis. Of these, dermal hyalinosis is a hallmark of both diseases.

**Figure 4 diagnostics-12-03070-f004:**
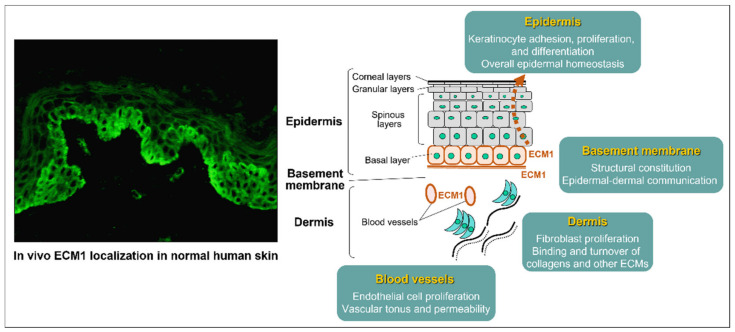
Multifocal interaction of in vivo ECM1 with surrounding extracellular matrix and structural molecules in the skin. ECM1 is expressed in the major skin components (particularly the epidermal basal layer and basement membranes) and adjunct appendages (blood vessel walls and follicular epithelium), as immunostained in normal human skin (**left panel**), and regulates the different in vivo turnover and feedback productivity of binding partners as a ‘biological glue’, contributing to the integrity and maintenance of skin homeostasis (**right panel**).

**Figure 5 diagnostics-12-03070-f005:**
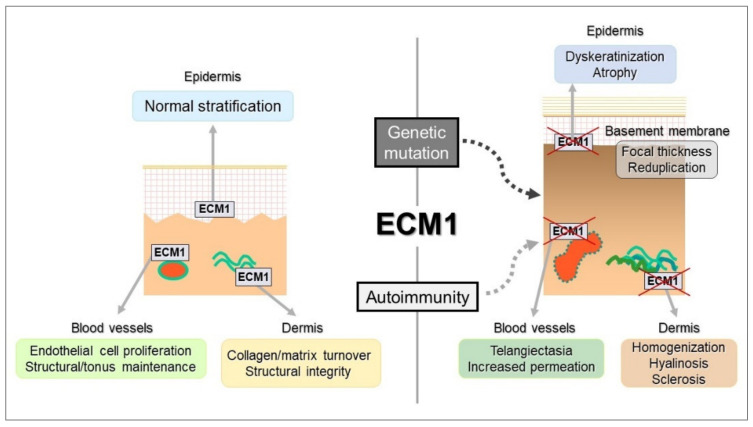
Schematic image for autoimmune and genetic impairment of ECM1 function in the skin. Genetic ablation and autoantibody targeting of the skin ECM1 cause dysregulation of its binding partners, as listed in Figure 3, contributing to the homeostatic imbalance or collapse in the epidermis (dyskeratosis and atrophy), dermis (collagen homogenization and sclerosis), and blood vessels (telangiectasia and impermeability). The chain of these statements finally establishes the pathological features seen in LiP and LS.

## Data Availability

Not applicable.

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
