# Peer review of "Lichen Sclerosus: A Current Landscape of Autoimmune and Genetic Interplay"

_diagnostics, 2022, doi:10.3390/diagnostics12123070_

Round 1

Reviewer 1 Report

The manuscript by Noritaka Oyama and Minoru Hasegawa is well written and very informative. The authors provide an excellent overview on the current understanding of lichen sclerosus. Further improvement may be possible in the display and description of histology and immunohistochemical analysis of lesions, but this is not essential.

Author Response

To the reviewer 1:

Thank you so much for your kind and specific advice that is helpful to further improve the manuscript. The reviewer kindly commented not all the suggestions need to be incorporated at this stage, but at least I have tried to change the description for the skin pathology more precisely and profoundly to provide information on how the pathological features of lichen sclerosus and a counterpart genodermatosis lipoid proteinosis are almost compatible [legend of the new Fig. 3 (previous Fig. 2), page 3, lines 139-141]. Also, the same statement has been reflected in the legend of the newly added skin pathology in Fig1.

As suggested, it is of great interest to know how much expression level of ECM1, as well as its binding partners, remains to be left in the LS lesional skin by immunohistochemical analysis. Also, this is true of their localization in the skin, which might be in part affected by impaired ECM1 function. These studies will further assist the underlying scenario surrounding the binding core ECM1, and surely need to be implemented in our next project.

English grammar and typing errors throughout the text have been corrected properly by a native English scientist.

Reviewer 2 Report

The manuscript entitled “Lichen sclerosus: a current landscape of autoimmune and genetic interplay” represents a review of unquestionable interest both in terms of the subject matter and practical accessibility.

Since this is a review, I suggest that the authors give more scope to the introductory section by delving into the etio-pathogenic hypotheses of LS, which are numerous and difficult to explain. They could also produce a table illustrating them in an explanatory and effective manner if they consider it appropriate.

About the other sections in which the argumentation of the review is developed, I believe that the authors have done a good job deepening the topics and also supplementing the text with up-to-date references.

I would suggest to the authors, if possible, to add more images of both macroscopic and histological lesions and to improve the definition of figure 3, which appears too dark and difficult to read. One possibility would be to make it similar to figure 4, also for a kind of parallelism between the figures in the manuscript.

With the exception of these minor corrections, I consider that the manuscript does not require any further editing in order to be published.

Author Response

To the reviewer 2:

Firstly, thank you so much for your kind and practical comments. All of these comments are indeed useful for further improving the manuscript and need to be incorporated. Please find our point-by-point responses to the comments as follows.

[Effective introduction for etiopathology]

According to the comment, we have tried to add more profound and specific explanations for extending the etiopathogenic evidence of LS in the introduction section (pages 1-2, lines 41-66), although I gave up adding a new explanatory table. I appreciate if you could kindly accept this.

[Macroscopic and microscopic figures]

As suggested, we have added both of these pictures; the former macroscopic pictures consisted of ‘extra-genital’ LS other than genital LS, particularly providing that the disease may occur at any skin sites and alert the differential diagnosis in the setting of clinical practice (new Fig. 2). The latter microscopic pictures consisted of those from female vulval LS skin, corresponding to the different clinical stages, which have been aligned in the lower panels of the new Fig. 3 (previous Fig. 2). Also, the legends of these newly added figures have been updated (page 2, lines 73-84/ page 3, lines 104-105).

[Poor figure 3]

As suggested, we have replaced the original Fig. 3 with the new version with an as similar schematic presentation to the new Fig. 5 (previous Fig. 4) as possible. Along with the replacement, the legend description has been changed properly (legend, page 6, lines 195-200).